

# Foliar spraying of indoleacetic acid (IAA) enhances the phytostabilization of Pb in naturally tolerant ryegrass by limiting the root-to-shoot transfer of Pb and improving plant growth

Chengqiang Zhu[1], Runhai Jiang[1], Shaofu Wen[1], Tiyuan Xia[1], Saiyong Zhu[2,3] and Xiuli Hou[1]

[1] Kunming University, Kunming, China
[2] Zhejiang Ecological Civilization Academy, Huzhou, China
[3] College of Environmental & Resource Sciences of Zhejiang University, Hangzhou, China

Corresponding authors
Saiyong Zhu, 0622411@zju.edu.cn
Xiuli Hou, hxlyn@aliyun.com

## ABSTRACT

Exogenous addition of IAA has the potential to improve the metal tolerance and phytostabilization of plants, but these effects have not been systematically investigated in naturally tolerant plants. Ryegrass (*Lolium perenne* L.) is a typical indigenous plant in the Lanping Pb/Zn mining area with high adaptability. This study investigated the phytostabilization ability and Pb tolerance mechanism of ryegrass in response to Pb, with or without foliar spraying of 0.1 mmol L$^{-1}$ IAA. The results indicated that appropriate IAA treatment could be used to enhance the phytostabilization efficiency of naturally tolerant plants. Foliar spraying of IAA increased the aboveground and belowground biomass of ryegrass and improved root Pb phytostabilization. Compared to Pb-treated plants without exogenous IAA addition, Pb concentration in the shoots of ryegrass significantly decreased, then increased in the roots after the foliar spraying of IAA. In the 1,000 mg kg$^{-1}$ Pb-treated plants, Pb concentration in the shoots decreased by 69.9% and increased by 79.1% in the roots after IAA treatment. IAA improved plant growth, especially in soils with higher Pb concentration. Foliar spraying of IAA increased shoot biomass by 35.9% and root biomass by 109.4% in 1,000 mg kg$^{-1}$ Pb-treated plants, and increased shoot biomass by 196.5% and root biomass by 71.5% in 2,000 mg kg$^{-1}$ Pb-treated plants. In addition, Pb stress significantly decreased the content of photosynthetic pigments and anti-oxidase activities in ryegrass, while foliar spraying of IAA remedied these negative impacts. In summary, foliar spraying of IAA could increase the biomass and improve the Pb tolerance of ryegrass.

## INTRODUCTION

Heavy metal contamination of soils has become an increasing concern worldwide (*Tripathi et al., 2021*). Yunnan Province is located in southwest China, and is rich in nonferrous metal resources. Due to the long history of mining and smelting in the area, soils of

many regions in Yunnan Province have been contaminated by heavy metals. For example, the Pb concentration range in the Lanping Pb/Zn mining area is 58.93~31,2451.54 mg kg$^{-1}$, with an average concentration of 9,993.31 mg kg$^{-1}$ (the screening level in China for building land is 800 mg kg$^{-1}$, GB 36600-2018; *Ruipin et al., 2009*). Because heavy metals do not degrade, heavy metal contamination is a perennial problem in soils and a threat to the surrounding environment (*Dhaliwal et al., 2020*). Plant growth and development are impacted by Pb-contaminated soils. Plants under Pb stress show symptoms of oxidative stress and photosynthesis inhibition (*Chaki, Begara-Morales & Barroso, 2020*; *Cid, Pignata & Rodriguez, 2020*; *Xiang et al., 2021*), which affect plant growth and survival. Phytostabilization through locally-grown, tolerant plant species could be a potential solution for controlling Pb soil contamination, but phytostabilization efficiency is largely restricted in seriously contaminated soils due to inhibited plant growth. Investigating the Pb tolerance mechanisms of plants and improving plant tolerance to Pb are important for effectively using phytostabilization to restore Pb-contaminated soils.

Oxidative stress is the most common toxic mechanism that hinders plant growth and colonization in metal-contaminated soils. In stressed environments, excessive reactive oxygen species (ROS) are generated and accumulate in plant tissue, resulting in irreversible oxidative damage to functional molecules like lipids, proteins, DNA and chloroplast pigments (*Chaki, Begara-Morales & Barroso, 2020*; *Gill & Tuteja, 2010*; *Li et al., 2018*). Plant growth and development depend largely on the assimilates produced from photosynthesis, and the photosynthetic system is very vulnerable to Pb. Pb stress inhibits seed germination, restrains root development, impairs the photosynthetic system and reduces biomass accumulation (*Cid, Pignata & Rodriguez, 2020*; *Xiang et al., 2021*). Reduced photosynthetic pigment content and weakened chlorophyll fluorescence intensity have been observed in Pb-stressed environments. *Tan et al. (2022)* reported reduced photoresponse of the PS II system and reduced chlorophyll content in cabbage under Pb stress, and similar trends were also observed by *Xie, Pu & Xiong (2021)*.

Many studies have shown that metal stress leads to negative effects on the growth and physiological activities of plants, but some studies have also observed up-regulated physiological activities under stressed environments, called the "stimulatory effect" (*Costantini, Metcalfe & Monaghan, 2010*; *Jia et al., 2013*). The stimulatory effect is an adaptive evolutionary response that is directly regulated by phytohormones. The interactions between phytohormones lead to the up- or down-regulation of stress-response genes and their regulatory factors, remedying the physiological injury provoked by metal stress (*Bucker-Neto et al., 2017*). Indoleacetic acid (IAA) is the most common natural auxin that participates in plant growth and development as well as the defense against environmental stress.

Some basic physiological activities such as root elongation, gravitropism, root hair development and lateral root development, are mainly regulated by IAA (*Anfang & Shani, 2021*; *Luo, Zhou & Zhang, 2018*; *Pei et al., 2010*). Adaptive fluctuations in endogenous IAA level have been observed in stressed plants. Fe$^{2+}$ deficiency led to a significant increase in root IAA levels in *Arabidopsis thaliana* (*Chen et al., 2010*). In a saline-stressed environment, tomato (*Lycopersicon esculentum*) roots produced significantly higher IAA and the shoots

produced less IAA compared to tomato plants growing in a non-stressed environment (*Albacete et al., 2008*). Compared to non-stressed plants, IAA content increased in a low saline-stressed environment and decreased in a high saline stressed-environment (*Shao et al., 2016*). *Takshak & Agrawal (2017)* found that exogenous IAA supplementation could remedy the oxidative damage provoked by ROS and enhance the abiotic stress resistance of plants. In addition, exogenous IAA supplementation increased photosynthetic pigment content and activated the xanthophyll cycle, which helps protect photosynthetic organs against various environmental stresses (*Alicja et al., 2018*). Other auxin analogues demonstrated similar effects. Exogenous naphthylacetic acid (NAA) increased the content of semicellulase in the roots of *Arabidopsis thaliana*, enhanced root phytostabilization and alleviated the Cd toxicity in the aboveground plant tissues (*Zhu et al., 2013*). Thus, exogenous auxin supplementation could be an effective way for enhancing plant tolerance to heavy metals and improving the phytostabilization ability of roots. However, the effects of exogenous IAA on naturally tolerant plant species have not been systematically investigated.

Ryegrass (*Lolium perenne* L.) is a typical indigenous plant in the Lanping Pb/Zn mining area with high adaptability, rapid growth rate and large biomass, which serves as an ideal phytostabilization material for restoring the contaminated soils (*Yan et al., 2011*). Using local tolerant plants for phytostabilization could be an economic and environmentally-friendly method of controlling the ecological risks of heavy metals (*Muthusaravanan et al., 2018*). However, the impact of metal toxicity in natural lead-resistant ryegrass and the response mechanism these plants use to alleviate this toxicity are still unclear. There are no data on metal toxicity mitigation and protection using IAA on the growth of natural lead resistant ryegrass. We hypothesized that exogenous IAA could improve the growth of natural lead-resistant ryegrass by limiting the transfer of Pb from roots to shoots, alleviating oxidative stress and protecting the photosynthetic system. Indigenous ryegrass in the Lanping Pb/Zn mining area was chosen to test this hypothesis. The present study aimed at investigating the following: (1) physiological changes in the photosynthetic system and anti-oxidase system, (2) the relationship between Pb accumulation and endogenous IAA level, and (3) root morphological changes in response to Pb stress. This study will aid in an understanding of the regulatory effect of IAA on the Pb tolerance of plants, and provide a scientific basis for using tolerant plants in soil remediation and vegetation restoration in heavy metal-polluted soils.

## MATERIALS & METHODS
### Experimental design
Seeds of Pb-tolerant ryegrass (*Lolium perenne* L.) were collected from the Jinding lead-zinc mining area, Lanping County, Nujiang City, Yunnan Province, where the Pb soil concentration reached 9,207.4 mg kg$^{-1}$ (collected from 26°46′N, 99°48′W), which is close to the average Pb concentration (9,993.31 mg kg$^{-1}$) of this region, as indicated in a previous study (*Ruipin et al., 2009*). A pot culture experiment was carried out in the greenhouse of the Academy of Kunming (24°98′N, 102°80′W). Soils used in the pot culture experiment
**Table 1 Soil properties (background values).**

| Soil properties | pH | Available P mg kg$^{-1}$ | Available K mg kg$^{-1}$ | Pb mg kg$^{-1}$ | Alkali-hydrolyzed nitrogen mg kg$^{-1}$ |
|---|---|---|---|---|---|
| value | $6.49 \pm 0.2$ | $82.1 \pm 5$ | $12.44 \pm 0.9$ | $110.4 \pm 10.2$ | $103.3 \pm 4.5$ |

were collected from Guanwu Mountain in the University of Kunming, Kunming, Yunnan (24°97′N, 102°81′W) with the soil background values shown in Table 1. The average temperature during plant growth was 25/15 °C in the day/night, and the average air humidity was 41%. Pots with a height of 16.5 cm and a diameter of 15.7 cm were used in the present study and each pot was filled with 2.5 kg of soil. Pb was amended as $Pb(NO_3)_2$ at the rate of 0 (CK), 250, 500, 1,000, 1,500 and 2,000 mg kg$^{-1}$. The amended soils were homogenized and left for incubation for 30 days before seeding. Fifteen uniform seeds were planted in each pot. An IAA concentration of 0.1 mmol L$^{-1}$ was selected for foliar spraying, as previous research indicated this concentration could significantly promote the growth of ryegrass plants (*Ran et al., 2020*). After 70 days of plant growth, the IAA treatments were applied by foliar spraying of 6 mL 0.1 mmol L$^{-1}$ IAA, while the control was sprayed with 6 mL water. Foliar spraying was performed every two days and repeated three times. All treatments were performed in four replicates.

## Sampling

The plants were harvested after 100 days of growth and separated into roots and shoots. The roots of ryegrass were soaked in 20 mmol L$^{-1}$ EDTA-Na$_2$ solution for 0.5 h and rinsed several times with distilled water to remove the adhering Pb.

## Sample analysis
### *Metal analysis*

The plant samples were oven-dried, ground and sieved (<2 mm) before analysis. Soil samples and Pb standard solution (GBW07445) were air dried and passed through two mm mesh before analysis. Leaf samples were subjected to mixed acid digestion (HNO3-HClO4-HF). Inductively coupled plasma mass spectrometry (ICP-MS, NexION 300X; PerkinElmer, Waltham, MA, USA) was used to determine the concentration of Pb.

## SOD and POD activity

The xanthine oxidase method was used to determine superoxide dismutase (SOD) activity. A total of 0.2 g of a fresh leaf or root was ground in pre-cooled mortar with the addition of extractant. After grounding, the mortar was rinsed with extractant (PB, Phosphate Buffer PH: 7.0) several times. The mixture was then centrifuged at 10,000 rpm for 15 min and the supernatant was collected for analysis. A commercial kit (Nanjing Jiancheng Bioengineering Institute) was used to determine SOD activity by measuring the light absorption value at 560 nm.

Peroxidase (POD) activity was measured using the colorimetric method. Tissue homogenate was prepared by mixing 0.2 g of fresh plant tissue with 1.8 mL phosphate buffer (pH = 7.0). After centrifuging at 3,500 rpm for 10 min, the supernatant was collected. A

commercial kit (Nanjing Jiancheng Bioengineering Institute) was used to determine POD activity by measuring the light absorption value at 420 nm.

## Photosynthetic indexes

Four seedlings were randomly selected in each treatment and 0.2 g of a fresh leaf was collected to determine the photosynthetic pigments. Photosynthetic pigments were extracted by 95% ethanol. The concentration of chlorophyll a (Ca), chlorophyll b (Cb) and carotenoid in the extractant was measured by light absorption at 663 nm, 646 nm and 470 nm, respectively, and calculated using the following equations:

Ca conc. $= (13.95 \times A665 - 6.88 \times A649) \times 20 \times 5/1000$

Cb conc. $= (24.96 \times A649 - 7.32 \times A665) \times 20 \times 5/1000$

Carotenoid conc. $= (1000 \times A470 - 2.05Ca - 114.8Cb)/245 \times 20 \times 5/1000$

where A665, A649 and A470 indicate the light absorption value of the leaf extractant at 665 nm, 649 nm and 470 nm, respectively.

The PS II effective photochemical quantum yield was measured at 09:00~11:00 am on December 15th 2022 using a portable modulated chlorophyll fluorescence apparatus (PAM-2500; Heinz Walz GmbH, Effeltrich, Germany).

## Endogenous IAA

The fresh leaves and roots were rinsed several times with distilled water to remove the adhering exogenous IAA. Then, the endogenous IAA concentration was measured using the ELISA method (kit was purchased from FANKEWEI company) with a microplate reader (SpectramaxABSplus; Molecular Devices, San Jose, CA, USA).

## Biomass measurement

The collected roots were rinsed with tap water to remove the adhering soil particles and then the root parameters, including total root length and fibrous root length, were analyzed using an LA-S root scanner and analyzer (Wanshen Technological Co., Hangzhou, China). For biomass measurement, the roots and shoots of ryegrass plants were collected, oven-dried, and weighed.

## Statistical analysis

Statistical analyses were performed using SPSS software packages (v.21.0; SPSS Inc., Chicago, IL, USA) and Origin 2022. All data were expressed as means plus or minus one standard error ($n = 4$). Two-way ANOVA was used to compare the differences between groups at the 0.05 confidence level.

# RESULTS

## Effects of exogenous IAA on Pb accumulation in Pb-stressed ryegrass

The Pb was mostly concentrated in the roots of ryegrass grown in the pot experiment. As the soil Pb level increased, the plant Pb concentration in roots and shoots generally increased with or without IAA (Fig. 1). Pb concentrations in the shoots of Pb-treated plants without IAA addition increased from 40 mg kg$^{-1}$ to 78 mg kg$^{-1}$ (Fig. 1A) as the soil Pb

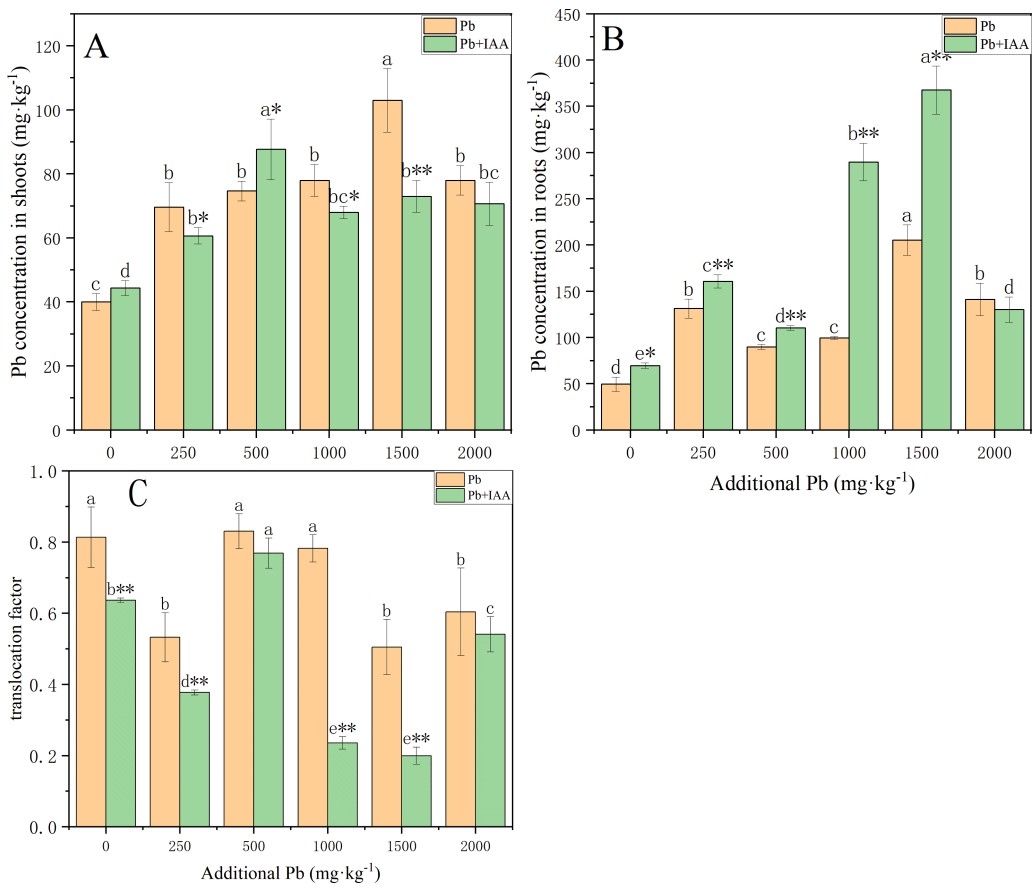

**Figure 1** **Pb concentration in shoots (A) and roots (B) with or without foliar spraying of IAA, and (C) the translocation factor of ryegrass.** Different letters indicate significant differences between different Pb concentrations at a 0.05 confidence level; asterisks (* and **) indicate significant differences between Pb and Pb+IAA groups at 0.05 and 0.01 significance levels, respectively.

concentration increased from 0 mg kg$^{-1}$ to 1,500 mg kg$^{-1}$, while the root Pb concentration increased from 49.7 mg kg$^{-1}$ to 140.5 mg kg$^{-1}$ (Fig. 1B). Foliar spraying of IAA generally inhibited the accumulation of Pb in the shoots. However, the Pb concentration in the shoots of plants growing in 500 mg kg$^{-1}$ Pb-spiked soil with IAA addition reached 94.7 mg kg$^{-1}$, which was higher than the plants without IAA addition. Pb translocation factors generally decreased as a result of foliar IAA spraying. The translocation factor of IAA+Pb-treated plants was significantly lower than that of Pb-treated plants at soil Pb concentrations of 0, 250, 1,000 and 1,500 mg kg$^{-1}$. Foliar IAA spraying reduced the Pb translocation factor to 30.1% of Pb-treated plants at the Pb concentration of 1,000 mg kg$^{-1}$.

## Effects of exogenous IAA on SOD and POD activity

Measurements of the effects of Pb and foliar spraying of IAA on SOD and POD activity, as shown in Fig. 2, indicated that IAA alleviated the oxidative stress provoked by soil Pb.

SOD activity in the shoots and roots of the Pb group (Fig. 2A and 2B) increased as soil Pb concentration increased from 0 mg kg$^{-1}$ to 500 mg kg$^{-1}$, peaked in 500 mg kg$^{-1}$ soil Pb,

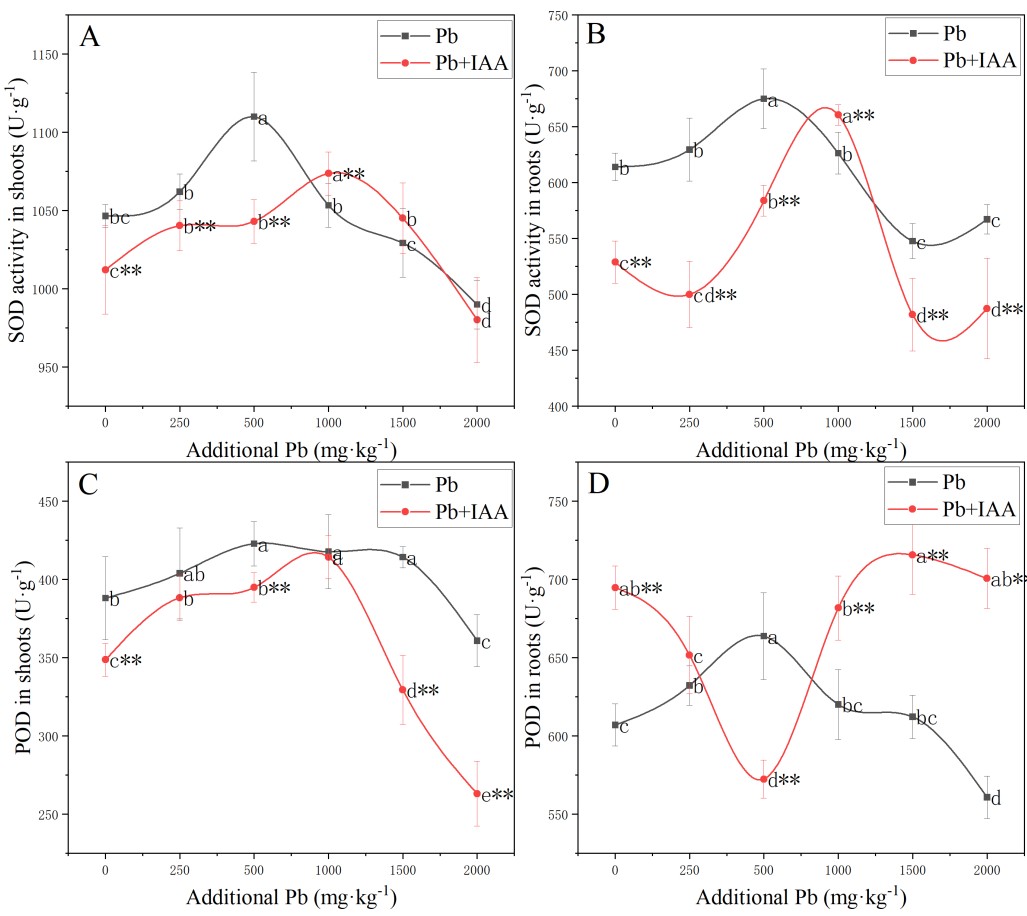

**Figure 2  Shoot SOD activity (A), root SOD activity (B), shoot POD activity (C) and root POD activity (D) of ryegrass growing in Pb-spiked soil with or without foliar spraying of IAA.** Different letters in the group indicate significant differences between Pb concentrations at a 0.05 confidence level; asterisks (* and **) indicate significant differences between the IAA+Pb and Pb groups at 0.05 and 0.01 significance levels, respectively.

and then decreased. Similarly, SOD activity in the shoots and roots of Pb+IAA group also followed a parabolic trend except that maximum SOD activity was recorded at 1,000 mg kg$^{-1}$ soil Pb. SOD activity was generally higher in the Pb group, indicating more serious oxidative stress in this group.

In the Pb treatment group, POD activity in the shoots (Fig. 2C) and roots (Fig. 2D) increased as soil Pb concentration increased from 0 mg kg$^{-1}$ to 500 mg kg$^{-1}$, peaking in 500 mg kg$^{-1}$ soil Pb. In the Pb+IAA group, POD activity in the shoots followed a similar parabolic trend, but was generally lower than that in the Pb group. POD activity in the roots (Fig. 2D) of the Pb+IAA group was generally higher than that in the Pb group and followed a reversed parabolic trend. POD activity in the roots of Pb+IAA-treated plants decreased in low soil Pb concentration, reached a minimum value at a soil Pb concentration of 500 mg kg$^{-1}$ (86.2% of that in Pb group at 500 mg kg$^{-1}$ Pb), and then increased. The
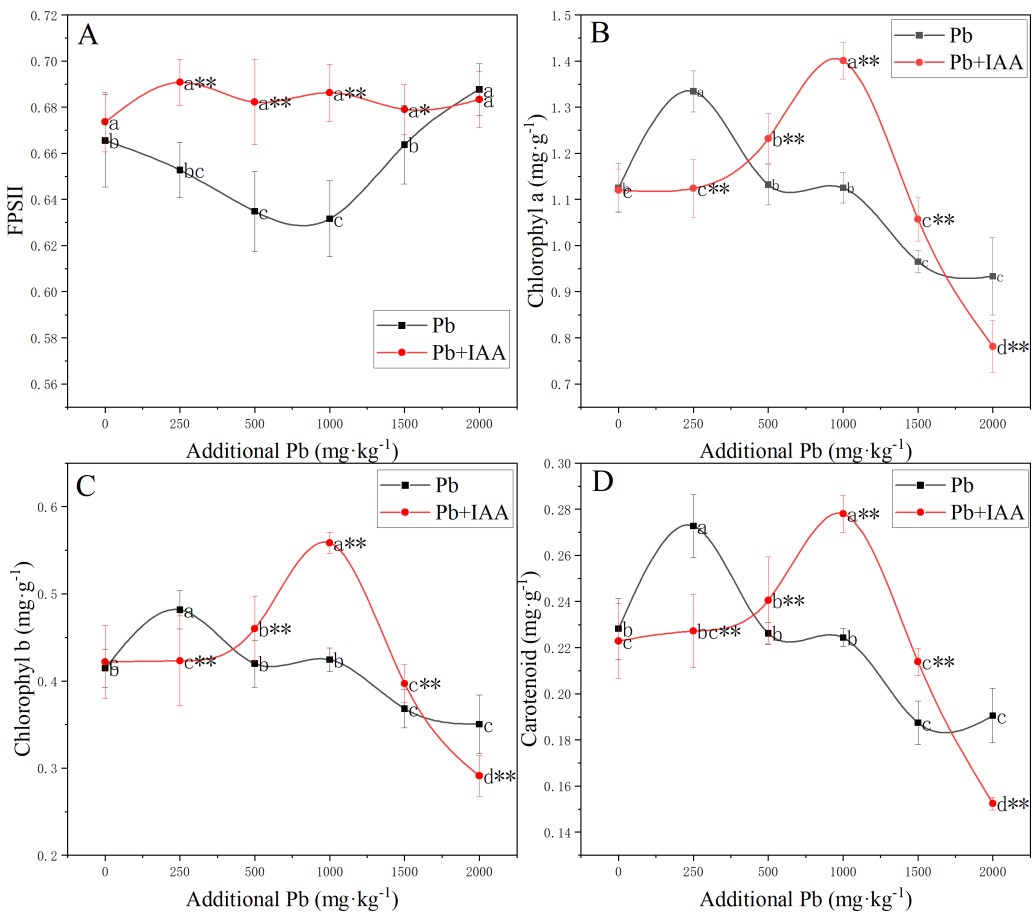

**Figure 3** (A–D) **Effective photochemical quantum yield of PSII and content of photosynthetic pigments in ryegrass under Pb stress with or without IAA.** Different letters in the group indicate significant differences between Pb concentrations at a 0.05 confidence level; asterisks (* and **) indicate significant differences between the Pb and Pb+IAA groups at 0.05 and 0.01 significance levels, respectively.

maximum POD activity value in the roots of the Pb+IAA group was recorded at 1,500 mg kg$^{-1}$ soil Pb concentration, which was 116.9% of the POD activity level in the Pb group.

## Effects of exogenous IAA on the photosynthetic system

The photosynthetic system was significantly affected by Pb and foliar spraying of IAA (Fig. 3). The photochemical efficiency of photosystem II (FPSII) in the Pb group was generally lower than that in the Pb+IAA group. The minimum value of FPSII in the Pb group was recorded at 1,000 mg kg$^{-1}$ soil Pb concentration, which was only 90.3% of that in the Pb+IAA group (Fig. 3A). Foliar spraying of IAA remedied this reduction of FPSII and the FPSII remained relatively stable regardless of soil Pb concentrations.

In Pb-treated plants, the content of photosynthetic pigments generally decreased as the soil Pb concentration increased, except that the contents of photosynthetic pigments were higher in plants growing in soils spiked with 250 mg kg$^{-1}$ Pb. Chlorophyll *a* (Fig. 3B), chlorophyll *b* (Fig. 3C) and carotenoid (Fig. 3D) contents in the 250 mg kg$^{-1}$ Pb group
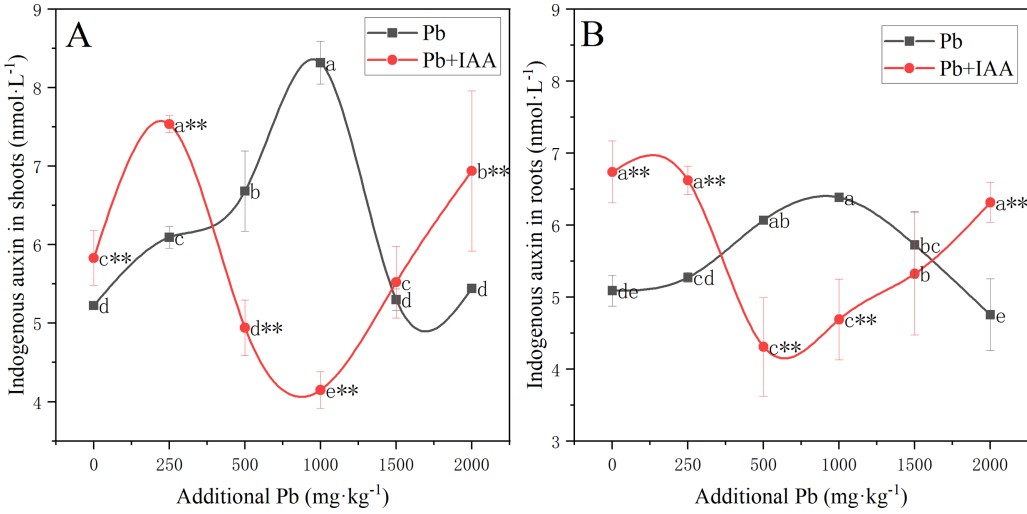

**Figure 4** **Endogenous IAA in the shoots (A) and roots (B) of ryegrass in Pb and Pb+IAA-treated plants.**
Different letters in the group indicate significant differences between Pb concentrations at a 0.05 confidence level; asterisks (* and **) indicate indicate significant differences between Pb and Pb+IAA groups at 0.05 and 0.01 significance levels, respectively.

were 118.7%, 113.9% and 120% of those in the Pb+IAA group, respectively. In the 1,000 mg kg$^{-1}$ Pb group, chlorophyll $a$, chlorophyll $b$ and carotenoid contents were only 80.3% (Fig. 3B), 76% (Fig. 3C) and 80.7% (Fig. 3D) of those in the Pb+IAA group, respectively. These results indicated that the foliar spraying of IAA could remedy photosynthetic system injuries induced by Pb stress.

## Effects of exogenous IAA on the level of endogenous IAA

The results of the effects of Pb and IAA on endogenous IAA level was shown in Fig. 4. In the Pb group, endogenous IAA in shoots (Fig. 4A) and roots (Fig. 4B) increased as the soil Pb concentration increased from 0 mg kg$^{-1}$ to 1,000 mg kg$^{-1}$. Endogenous IAA content in the Pb group was lower than that in the Pb+IAA group at Pb concentrations of 0 and 250 mg kg$^{-1}$, but was higher at Pb concentrations of 500, 1,000 and 1,500 mg kg$^{-1}$. Maximum endogenous IAA content was observed in the Pb group at Pb concentration of 1,000 mg kg$^{-1}$, which was 100.5% higher than the maximum endogenous IAA content observed in the IAA+Pb group in shoots (Fig. 4A) and 36.1% higher than that observed in the roots (Fig. 4B).

## The correlation between endogenous IAA and photosynthetic pigments or anti-oxidase activity

The correlations between the photosynthetic system, anti-oxidase activity and endogenous IAA were analyzed and the results are shown in Table 2. In the Pb-treated group, endogenous IAA in the shoots was positively correlated with chlorophyll $a$ ($p < 0.05$) and chlorophyll $b$ ($p < 0.01$), while endogenous IAA in the roots was positively correlated with POD activity ($p < 0.05$).

Zhu et al. (2023), *PeerJ*, DOI 10.7717/peerj.16560  9/19

**Table 2  The correlation between endogenous IAA and photosynthetic pigments or anti-oxidase activity.**

| Indogenous IAA | Chlorophyll *a* | Chlorophyll *b* | Carotenoid | Shoot SOD activity | Shoot POD activity | Root SOD activity | Root POD activity |
|---|---|---|---|---|---|---|---|
| Pb group roots | 0.29[*] | 0.345[**] | 0.233 | 0.233 | 0.22 | | |
| Pb group shoots | | | | | | −0.069 | 0.298[*] |
| Pb+IAA group shoots | −0.539[**] | −0.6[**] | −0.484[**] | −0.343[*] | −0.521[**] | | |
| Pb+IAA group roots | | | | | | −0.526[**] | 0.414[**] |

Notes.
[*,**] indicate significant correlation at 0.01 and 0.05 significance levels, respectively.

In the Pb+IAA-treated plants, endogenous IAA content in the shoots was positively correlated with chlorophyll *a* content ($p < 0.01$), chlorophyll *b* content ($p < 0.01$), caretenoid content ($p < 0.01$) and POD activity ($p < 0.01$), while the endogenous IAA content in the roots was positively correlated with POD activity ($p < 0.01$) and negatively correlated with SOD activity ($p < 0.01$).

### IAA improves the growth of *ryegrass* under lead stress

The effects of foliar IAA spraying on the biomass accumulation and root morphology of Pb-stressed plants are shown in Fig. 5. In Pb-treated plants without IAA addition, the biomass of the shoots and roots decreased as the soil Pb concentration increased. In the 2,000 mg kg$^{-1}$ Pb group, the shoot biomass, root biomass, root length and fibrous root length were reduced by 62.7% (Fig. 5A), 43.4% (Fig. 5B), 48.9% (Fig. 5C) and 59% (Fig. 5D), respectively, compared to those in the 0 mg kg$^{-1}$ Pb-treated plants. The ryegrass plants in the Pb+IAA group had significantly higher biomass compared to Pb-treated plants, especially at 1000 mg kg$^{-1}$ Pb concentration (root biomass increased by 109.4%, Fig. 5B) and 2,000 mg kg$^{-1}$ Pb concentration (shoot biomass increased by 196.5%). Similar trends were observed in fibrous root length and total root length. Figures 5E and 5F demonstrate that the roots of ryegrass growing in Pb-spiked soils with IAA addition (left) were much larger than those without IAA addition (right). These results indicate that exogenous IAA could remedy the inhibition effects of Pb on ryegrass growth.

### DISCUSSION

Pb concentration in the shoots was generally lower in the Pb+IAA-treated ryegrass plants compared to Pb-treated plants, while the Pb concentration in the roots was higher (Figs. 1A and 1B). *Tammam et al. (2021)* also reported increased root Pb concentration in *Glebionis coronaria* L. after IAA addition. Pb is a non-essential element for plants and it competes with Ca$^{2+}$ for the membrane Ca$^{2+}$ channel to enter into the xylem vessel through the symplast pathway (*Huang & Cunningham, 1996*; *Moon et al., 2019*). Foliar auxin spraying has been shown to modulate the gene expression of membrane proteins, including Ca$^{2+}$ channel protein. *Zhu et al. (2013)* reported the down-regulated expression of IRT1, ZIP1, ZIP3 and ZIP4 after NAA application, which led to reduced transfer of Pb from the roots to shoots in *Arabidopsis thaliana*. Results of the present study demonstrated that foliar IAA spraying improved root growth, as indicated by increased root biomass and root

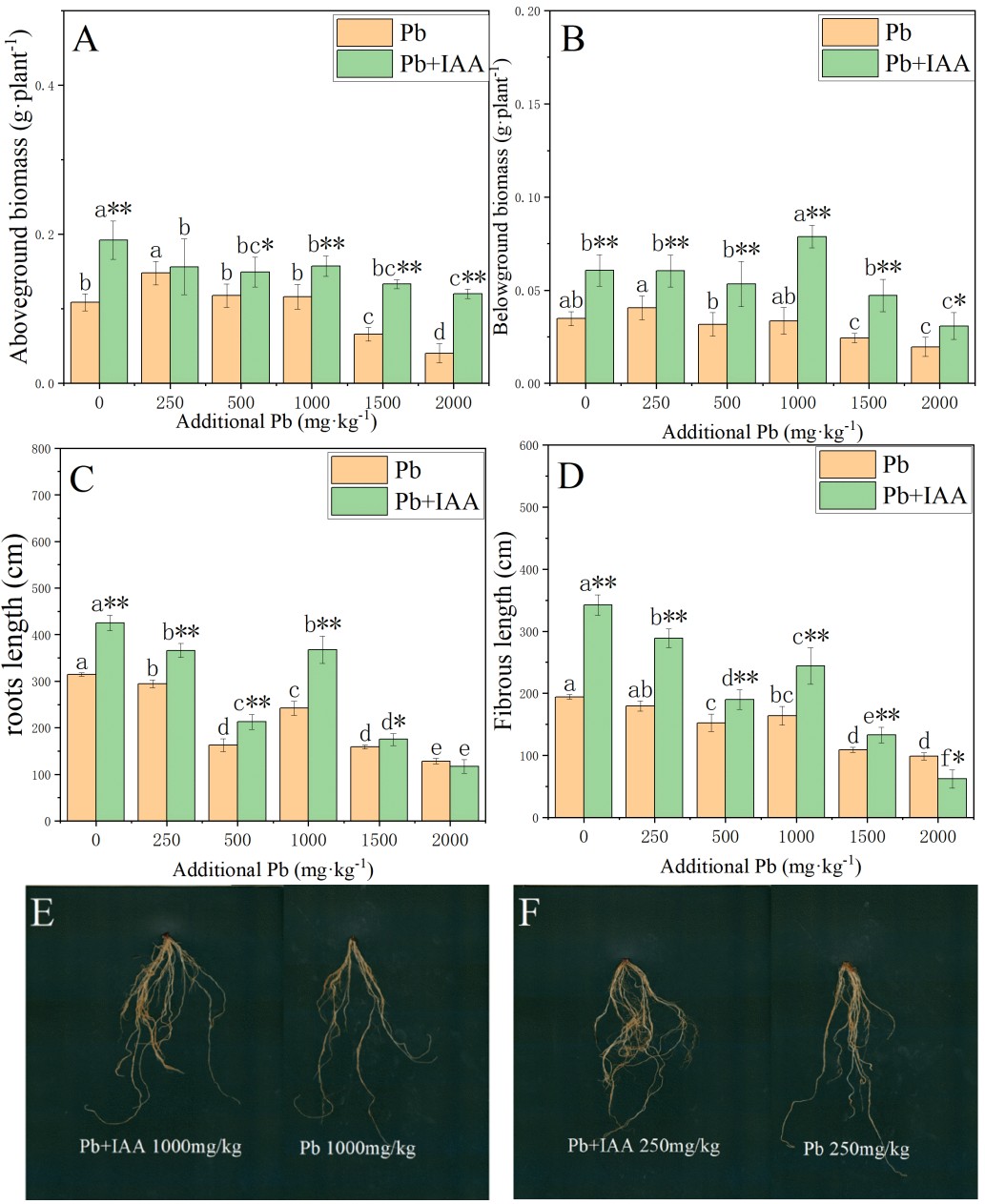

**Figure 5** **Shoot biomass (A), root biomass (B), total root length (C), fibrous root length (D) in Pb and Pb+IAA treated plants. E and F presented the roots of Pb+IAA (left) and Pb (right) treated plants.** Different letters in the group indicate significant differences between Pb concentrations at a 0.05 confidence level; asterisks (* and **) indicate significant differences between Pb and Pb+IAA groups at 0.05 and 0.01 significance levels, respectively.

length (Fig. 5). This improved root growth would increase the rhizosphere area of roots, potentially enhancing the plant uptake of soil Pb (*Yu et al., 2021*).

Plant growth and biomass accumulation are directly correlated with photosynthesis, the source of energy for plant development. Photosynthetic pigments are indispensable

elements of the photosynthetic system (*Zhang et al., 2018*). Carotenoid is also a natural antioxidant (*Simkin et al., 2022*). The PSII-effective photochemical quantum yield remains stable in non-stressed plants, but decreases significantly in stressed plants (*Lu et al., 2019*). In the Pb-treated plants without IAA addition in this study, the contents of photosynthetic pigments in ryegrass (*Lolium perenne* L.) generally decreased as the soil Pb concentration increased, except that the 250 mg kg$^{-1}$ Pb stimulated photosynthetic pigment production (Fig. 3). *Yang et al. (2020)* also reported similar trends in the variation of photosynthetic pigment content in response to Pb in *Davidia involucrata*. The ryegrass seeds used in this study were locally grown in the Lanping Pb/Zn mining area, and are naturally tolerant to Pb. The observed increase in the contents of photosynthetic pigments in 250 mg kg$^{-1}$ Pb concentration might be a result of the stimulatory effect. In the IAA+Pb group, the contents of photosynthetic pigments were highest at a Pb concentration of 1,000 mg kg$^{-1}$ (Fig. 3). Photosynthetic pigment content in the Pb+IAA group was significantly higher than that in the Pb group, demonstrating the protective effect of IAA on photosynthetic pigments. Foliar IAA spraying reduced Pb concentration in the shoots (Fig. 1A), increased the effective photochemical quantum yield of PSII (Fig. 3A), and increased the chlorophyll content (Figs. 3B and 3C) and carotenoid content (Fig. 3D), which was in accordance with the results of a previous study (*Jie et al., 2019*), indicating the important role of IAA in protecting the photosynthetic system against Pb stress.

Heavy metals are inductive agents of lipid peroxidation, which generates excessive ROS, leading to oxidative damages in biomacromolecules and lipid peroxidation of the cell membrane (*Li et al., 2023*). As the scavengers of ROS, anti-oxidase activity, especially SOD and POD activity, serve as oxidative stress indicators. In the present study, low concentration of Pb induced an increase in anti-oxidase activity in the shoots and roots of ryegrass (with the highest activity recorded at 500 mg kg$^{-1}$ soil Pb) and high Pb concentration decreased anti-oxidase activity (Fig. 2), which was in accordance with the results of a previous study (*Li et al., 2018*).

In the Pb+IAA-treated plants, SOD activity was lower than that in the Pb-treated plants at Pb concentrations of 0 and 500 mg kg$^{-1}$, but SOD activity was higher in Pb-treated plants at a Pb concentration of 1,000 mg kg$^{-1}$ Pb (Figs. 2A and 2B). *Alicja et al. (2018)* reported a similar inhibition effect of exogenous IAA on anti-oxidase activity in low Pb concentrations. IAA might enhance the ability of ryegrass roots to compartmentalize Pb, decrease the transfer of Pb from roots to shoots and reduce oxidative stress in the shoots. In the Pb+IAA group, POD activity in the shoots was generally lower than that in the Pb group, while root POD activity was generally higher. A regression analysis demonstrated a significant correlation between anti-oxidase activity and endogenous IAA, indicating the modulating effects of IAA on the anti-oxidative system. IAA also serves as an antioxidant (*Simkin et al., 2022*) and exogenous IAA could also reduce ROS. These results indicate IAA can relieve the oxidative stress provoked by soil Pb.

In Pb-treated plants without IAA addition, the significant positive correlation observed between endogenous IAA and chlorophyll content indicates that IAA might favor the production and maintenance of chlorophyll. Moderate Pb stress (<1,000 mg/kg) increased IAA content, which is in accordance with the findings of *Malkowski et al. (2020)*. It has

been suggested that the cellular level of auxins increases under a low level of abiotic stress, which stimulates vegetative growth, leading to the stimulatory effect (*Shahid et al., 2019*). In this study, severe Pb stress (>1,000 mg kg$_{-1}$) reduced endogenous IAA content (Fig. 4). Previous studies have shown that the inhibition of IAA biosynthesis under Pb stress is a main contributor to homeostasis disturbance and growth inhibition in plants (*Makowski et al., 2020*), and Pb stress can inhibit IAA biosynthesis by increasing IAA oxidase activity (*Xu et al., 2010*).

In Pb+IAA-treated plants, root and shoot endogenous IAA content increased at high soil Pb concentrations (500–2,000 mg kg$^{-1}$, Figs. 4A and 4B), showing the protective effects of exogenous IAA on IAA biosynthesis under Pb stress. Carotenoid content significantly increased as a result of foliar IAA spraying, also indicating anti-oxidant activity (*Simkin et al., 2022*). Enhanced root stabilization reduced oxidative stress in ryegrass plants, which inhibited the activity of IAA oxidase. Another possible reason for increased IAA content could be the upregulation of IAA biosynthesis or the IBA-to-IAA conversion system by exogenous IAA (*Zhao et al., 2021*).

A low concentration of Pb (250 mg kg$^{-1}$) stimulated the photosynthetic system (Fig. 3), increased anti-oxidase activity (Fig. 2) and improved plant growth (Fig. 5), which might be due to the stimulatory effects and adaptive response of plants to their surrounding environments (*Costantini, Metcalfe & Monaghan, 2010*; *Jia et al., 2013*). Previous research has demonstrated plant growth inhibition when the soil Pb concentration is high, possibly due to the adsorption saturation of Pb by the cell wall, leading to more mobile Pb reaching the protoplasm (*Niu et al., 2012*; *Zhou et al., 2017*). Bahram et al. (*Gholinejad et al., 2020*) found that the content of photosynthetic pigments, biomass, seed vitality and emergence rate of non-tolerant ryegrass plants were negatively affected by a soil Pb concentration of 250 mg kg$^{-1}$. *Jin et al. (2016)* also reported reduced growth rate in non-tolerant ryegrass in response to 300 mg kg$^{-1}$ soil Pb concentration. In the present study, the stimulatory effects of 250 mg kg$^{-1}$ soil Pb concentration demonstrated that the ryegrass plants growing in Lanping Pb/Zn were Pb tolerant.

Many studies have shown that IAA improves root and shoot growth in metal-stressed plants (*Egamberdieva, 2009*; *Piotrowska-Niczyporuk et al., 2018*; *Zhao & Jiang, 2013*). This study found that IAA-treated plants possessed significantly higher root biomass, total root length and fibrous root length (Fig. 5). IAA can accelerate cell division, cell elongation and lateral root development (*Kircher & Schopfer, 2018*; *Pasternak et al., 2007*; *Procko et al., 2016*), resulting in improved root growth. *Ouzounidou & Ilias (2005)* demonstrated that IAA alleviated the toxicity of $Cu^{2+}$ in the roots of sunflower, as shown by increased root length and more root hairs. A similar result was also found in *Arabidopsis thaliana* (*Agami & Mohamed, 2013*). In summary, for naturally tolerant plant species, foliar spraying of IAA can improve plant tolerance to Pb, increase biomass production and enhance phytostabilization efficiency.

## CONCLUSIONS

This study showed that Pb accumulation in the aboveground part of ryegrass negatively affected the anti-oxidase system and photosynthetic system, leading to aggravated oxidative

stress, decreased photosynthetic efficiency and inhibited plant growth. Foliar spraying of IAA increased the root-stabilized Pb content and reduced the transfer of Pb from the roots to the shoots. In this way, foliar spraying of IAA mitigated metal toxicity in the shoots, relieved oxidative stress and protected the photosynthetic system. Foliar spraying of IAA can improve the Pb tolerance of naturally tolerant plant species and increase root phytostabilization efficiency. Locally-grown, naturally tolerant plants combined with foliar IAA spraying is an accessible option for restoring contaminated soils and reducing the risk of heavy metal migration to the surrounding environments.

### Funding

This work was supported by the National Natural Science Foundation of China(42167009, 31300349, U1902202), the Special Project of Basic Research in Yunnan Local Colleges and Universities (2018FH001-004), the International Joint Innovation Team for Yunnan Plateau Lakes and Great Lakes of North America which is sponsored by the Yunnan Provincial Education Department (XC) and the Major Program for Basic Research Project of Yunnan Province(202101BC070002). The funders had no role in study design, data collection and analysis, decision to publish, or preparation of the manuscript.

### Grant Disclosures

The following grant information was disclosed by the authors:
The National Natural Science Foundation of China: 42167009, 31300349, U1902202.
Special Project of Basic Research in Yunnan Local Colleges and Universities: 2018FH001-004.
International Joint Innovation Team for Yunnan Plateau Lakes and Great Lakes of North America which is sponsored by Yunnan Provincial Education Department (XC) and Major Program for Basic Research Project of Yunnan Province: 202101BC070002.

### Competing Interests

The authors declare there are no competing interests.

### Author Contributions

- Chengqiang Zhu conceived and designed the experiments, performed the experiments, analyzed the data, prepared figures and/or tables, authored or reviewed drafts of the article, and approved the final draft.
- Runhai Jiang performed the experiments, authored or reviewed drafts of the article, and approved the final draft.
- Shaofu Wen performed the experiments, authored or reviewed drafts of the article, and approved the final draft.
- Tiyuan Xia performed the experiments, authored or reviewed drafts of the article, and approved the final draft.

- Saiyong Zhu conceived and designed the experiments, performed the experiments, analyzed the data, authored or reviewed drafts of the article, and approved the final draft.
- Xiuli Hou conceived and designed the experiments, prepared figures and/or tables, authored or reviewed drafts of the article, and approved the final draft.

## Data Availability

The raw measurements are available in Supplemental Files.

## Supplemental Information

Supplemental information for this article can be found online at http://dx.doi.org/10.7717/peerj.16560#supplemental-information.

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
