# Peer review of "Foliar spraying of indoleacetic acid (IAA) enhances the phytostabilization of Pb in naturally tolerant ryegrass by limiting the root-to-shoot transfer of Pb and improving plant growth"

_PeerJ, doi:10.7717/peerj.16560_

## Round 0.1 · original submission · Major Revisions

I would also recommend using a proficient English Speaker to improve the English quality.

In addition, the statistical analyses should be improved throughout the manuscript.

Reviewer 1 ·

Basic reporting

The manuscript deals with the Foliar spraying of IAA enhances the phytostabilization of Pb in naturally tolerant ryegrass by improving the plant growth and limiting the root-to-shoot transfer of Pb. The topic is very interesting. Methodologies followed are standard. The objectives of the study are focused. The authors should improve the aim of the work.
Please clarify the novelty and the justification of your work in the text.

The language should be revised by professional or native language speaker. The manuscript would benefit from the service of a professional language editor.
Please follow the abbreviation rule. Try to write the full name in their first appearance and abbreviation within the bracket, then write the only abbreviation form. Maintain this throughout the manuscript. In some place, you wrote abbreviation in the first appearance without mentioning their full name.

References need to be improved in the same format according to the journal’s requirements.

Experimental design

In the materials and methods section, what is the basis of selection the concentration of 0.1 mmol L-1 IAA.
It would be better to provide data on the physical and chemical properties and element content of the experimental soil.
Please include the data of preliminary experiment of selecting the concentration of IAA.
What is the dimension of pots receiving 50 seedling?

The quality control data should be given.
Recovery means of certified reference materials should be given in the manuscript.

Validity of the findings

In all Figures enlarge font to be more visible.
Statistical analysis of Figures is wrong.

Additional comments

The Discussion section needs improve in order to become more clear -Discussion is further collection of reports. The reasons for results and relation between data enough has been not emphasized in this section. The manuscript describes Pb concentration,SOD activity , POD activity,effective photochemical quantum yield of PSII , photosynthetic pigments contents, endogenous IAA and biomass, but discussion concerning relation between these effects is very poor.
The discussion section should provide relevant information on IAA interacted with Pb stress.
The use of IAA alone cannot be taken to suggest that exogenous IAA was responsible for the protection against the Pb stress under study in this work. This is also the major weakness in this work.

Reviewer 2 ·

Basic reporting

The article needs to ne improved. The statistical analysis must be changed.
Two variables are present at the work at the same time: the various concentrations of Pb and the application or not of IAA. A one-way ANOVA cannot be applied in this case, analyzing the results as a whole.
A two-way ANOVA should be used comparing both the treatments within each Pb concentration, and comparing each treatment in the several Pb concentrations. The p values of the interactions should also be added to each figure or table.
The results and discussion sections must be rewritten after the new statistical analysis.

Experimental design

No comment.

Validity of the findings

The results and statistical analysis need to be improved.

Additional comments

I advise authors to redo the statistics of all the results obtained (Two-way ANOVA), and then rewrite the entire results and discussion section already with the support of the new analysis to be carried out. Only then can a more careful analysis be made.
In fact only after redoing the statistical analysis and rewriting the results and discussion will it be possible to rigorously evaluate the article.

Annotated reviews are not available for download in order to protect the identity of reviewers who chose to remain anonymous.

Reviewer 3 ·

Basic reporting

Current article describes the role of IAA in phytostabilizing Pb in maize. Topic of research is very important. However, the manuscript requires thorough revision in terms of its grammar. As, in overall draft, poor quality english language is used. I would also recommend using a proficient English Speaker to improve the english quality.
In addition, references used are too old, and require updating. Similarly, results section could be further improved by adding the data of bioconcentration and translocation factors as research theme is very relevant to these factors. Major improvement in the abstract and conclusion section is required. Therefore I reject this manuscript as it is not acceptable in its current form

Experimental design

Expermiental Design used is One way ANOVA, which in my opinion, is not a true one in present case.

Validity of the findings

Findings are somewhat valid and incomplete to some extent as in my view, phytostabilization or phytoextraction studies require data of bioconcentration and translocation factors and hence, their results should have been added in the results section.

Annotated reviews are not available for download in order to protect the identity of reviewers who chose to remain anonymous.

---

## Round 0.2 · accepted · Accept

I have confirmed with PeerJ staff that the items mentioned by the reviewer can be addressed during the proofing stage.

Reviewer 1 ·

Basic reporting

Please carefully corrected the grammatical, styling, and typos found in your manuscript.

Experimental design

no comment

Validity of the findings

no comment

Additional comments

References need to be improved in the same format according to the journal’s requirements.
There are duplications in some references. For example,
Li D, Zhang L, Chen M, He X, Li J, and An R. 2018a. Defense Mechanisms of Two Pioneer Submerged Plants during Their Optimal Performance Period in the Bioaccumulation of Lead: A Comparative Study. Int J Environ Res Public Health 15. 10.3390/ijerph15122844
Li D, Zhang L, Chen M, He X, Li J, and An R. 2018b. Defense Mechanisms of Two Pioneer Submerged Plants during Their Optimal Performance Period in the Bioaccumulation of Lead: A Comparative Study. International Journal of Environmental Research & Public Health 15:2844.

Makowski E, Sitko K, Micha S, aneta G, Pogrzeba M, Kalaji HM, and Zielenik-Rusinowska P. 2020. Hormesis in Plants: The Role of Oxidative Stress, Auxins and Photosynthesis in Corn Treated with Cd or Pb. International Journal of Molecular Sciences 21:6.
Malkowski E, Sitko K, Szopinski M, Gieron Z, Pogrzeba M, Kalaji HM, and Zieleznik-Rusinowska P. 2020. Hormesis in Plants: The Role of Oxidative Stress, Auxins and Photosynthesis in Corn Treated with Cd or Pb. Int J Mol Sci 21. 10.3390/ijms21062099

Please carefully corrected the grammar and format in the manuscript.

Reviewer 3 ·

Basic reporting

I am satisfied with the authors response

Experimental design

Experimental design seems good

Validity of the findings

Findings are valid in lieu of current scenario